# Barriers and Facilitators to the Elimination of Mother-to-Child Transmission Services Among Pregnant and Breastfeeding Women in Gauteng Province, South Africa

**DOI:** 10.3390/nursrep15090318

**Published:** 2025-09-02

**Authors:** Ndivhuwo Mukomafhedzi, Takalani Tshitangano, Shonisani Tshivhase

**Affiliations:** 1Department of Public Health, School of Health Sciences, University of Venda, Thohoyandou 0950, South Africa; 2Department of Public Health, Faculty of Health Care Sciences, University of Limpopo, Polokwane 0727, South Africa; takalani.tshitangano@ul.ac.za

**Keywords:** EMTCT services, HIV, poor utilization, strategies, vertical transmission

## Abstract

**Introduction**: Eliminating mother-to-child transmission (EMTCT) of HIV is a global health priority to ensure that no child is born with HIV. When EMTCT services are underutilized, mothers and babies face greater risks, including the vertical transmission of HIV and higher rates of maternal and neonatal mortality. Despite ongoing efforts, many women worldwide still struggle to access and use these vital services. **Objective**: This study sought to explore barriers and facilitators to the elimination of mother-to-child transmission services among pregnant and breastfeeding women (PBFW) in Gauteng province, South Africa. **Methods**: A qualitative, explorative, and descriptive research design was used. Convenience and purposive sampling were used to select participants. The study population consisted of PBFW aged 18 years or above who were utilizing EMTCT services. Data was collected through in-depth face-to-face individual interviews with participants. A semi-structured interview guide was used to collect data until data saturation was reached after interviewing 25 participants. Data were analyzed using thematic analysis (Tesch’s open coding method). Trustworthiness and ethical principles were ensured. **Results**: Four main themes emerged from the data analyzed, namely, barriers associated with EMTCT service utilization, facility-based strategies to improve EMTCT service uptake, community support for enhancing EMTCT engagement, and the role of partner support in service utilization, each with linked sub-themes. This study found that health education about EMTCT, along with community awareness and involvement, encourages the target group to utilize these services. **Conclusions**: Increasing women’s use of EMTCT services is an important step toward eliminating MTCT and increasing the health and well-being of mothers and their children. Addressing numerous barriers to receiving these services, as well as implementing targeted measures, can help ensure that all women gain access to the care and support that they require to safeguard their families from HIV.

## 1. Introduction

The elimination of mother-to-child transmission (EMTCT) of HIV is an essential global health objective, recommended by the World Health Organization (WHO) as part of efforts to end the HIV/AIDS epidemic by 2030 [1]. Worldwide, nearly 1.3 million women living with HIV become pregnant each year, and without effective interventions, transmission rates from mother to child can be as high as 45% [2]. The adoption of comprehensive EMTCT programs, including early antenatal care (ANC) booking, HIV testing, and timely initiation of antiretroviral therapy (ART), has significantly reduced the global burden of pediatric HIV to <1% [3]. As a result, several countries have made tremendous progress in integrating HIV therapy with mother and child health [3]. However, despite these advances in lowering new infections, an estimated 1.4 million children were living with HIV in 2023 [2].

South Africa has the largest HIV epidemic worldwide and has made substantial progress in scaling up EMTCT services, achieving increased ANC attendance and ART coverage among pregnant women [4,5]. Despite these advances, vertical transmission of HIV persists, driven by factors such as delayed ANC booking, poor adherence to ART, stigma, and socio-cultural barriers [6,7]. In response, the government has put in place several initiatives and programs that integrate HIV prevention, care, and treatment into maternal and child health [5,6]. Despite these initiatives, many women still face significant barriers hindering them from utilizing EMTCT services [7]. Studies have shown that women’s knowledge, attitudes, and support systems strongly influence their engagement with EMTCT services [8]. Moreover, issues such as gender-based violence, partner non-disclosure, and substance abuse further affect service utilization and retention in care [9,10].

These findings highlight the need to improve access to care for PBFW living with HIV. Stigma and discrimination from their communities, families, and even healthcare providers discourage them from seeking care and disclosing their HIV status [10]. Although the barriers to service utilization are extensively documented, there is a lack of understanding of the techniques that women perceive as effective in enhancing EMTCT adoption. Few studies have examined how women perceive practical, context-specific approaches to make information more accessible and acceptable, which is a significant gap in our understanding.

Understanding the barriers and facilitators affecting PBFW’s use of EMTCT services is essential for tailoring effective interventions that address these challenges in the South African context. Interventions informed by women’s experiences are more likely to encourage service utilization, reduce the risk of MTCT, and achieve the goal of an AIDS-free generation. This study aims to explore barriers and facilitators to the elimination of mother-to-child transmission services among PBFW in Gauteng province, South Africa.

## 2. Methods

A qualitative, explorative, and descriptive design was used to collect options and views of PBFW on the barriers and facilitators to improve utilization of EMTCT services in Gauteng province, South Africa.

### 2.1. Study Setting

This study was conducted in Ekurhuleni, Gauteng Province, South Africa, from 18 April 2023 to October 2023. The City of Ekurhuleni is the second-largest district in the eastern part of the province and is divided into three sub-districts supported by 93 fixed clinics and 6 hospitals. Around 700 nurses trained in NIMART provide EMTCT services, serving an estimated 8000 pregnant and breastfeeding women living with HIV each quarter. It was chosen because of its diverse population and high HIV prevalence, making it an essential context for understanding EMTCT service utilization [11].

### 2.2. Study Population and Sampling

The study population was PBFW in Gauteng Province, South Africa, who attended EMTCT services at selected facilities (Winnie Mandela, Esangweni CHC, Nokuthela Ngwenya CHC, Daveyton Main clinic, Phola Park CHC, and Kempton Park Civic Centre clinic). The target population consisted of women aged 18 years or above who were utilizing EMTCT services during the study period. A non-probability sampling method was explicitly used because it was convenient to select PBFW with the required characteristics of accessibility and availability [12]. Data saturation occurred after interviewing 25 participants through in-depth individual interviews.

### 2.3. Data Collection

Data was collected through in-depth interviews with PBFW utilizing EMTCT services. The languages preferred by participants to express themselves comfortably during data collection were English, isiZulu, Sepedi, Setswana, and Xitsonga. Two main open-ended questions guided the interview: Question 1: In your own opinion, what might be the reasons for you not utilizing EMTCT services? Question 2: What do you think should be done to improve EMTCT services utilization? This was followed by probing questions, which were asked based on the participant’s responses to gather detailed information. A semi-structured interview guide was used to allow participants to express themselves in their own words. The guide was piloted with 10% of the sample (*n* = 25) to ensure clarity, cultural relevance, and understanding, and adjustments were made before the main interviews. Written consent was obtained from all participants above the age of 18 years, and permission to use a voice recorder for the interview was obtained. Interviews were conducted in a private room for 30–45 min. Observations and field notes were also taken to provide additional context. Probing was used to encourage participants to elaborate on their experiences and share more insight [13].

### 2.4. Measure of Trustworthiness

To ensure trustworthiness, the study adopted Lincoln and Guba’s four criteria [14]. Credibility was established by double-checking audio tapes to ensure correct transcription and field notes taken during the interviews. Credibility was ensured through direct quotes from participants. Transferability was guaranteed by providing enough descriptive comments so that the readers to assess the applicability of findings in other settings. Dependability was maintained by consulting an independent researcher to review the transcript and develop themes until a consensus was established. To ensure confirmability, two participants were contacted to represent other patients to make sure that the themes adequately reflected what participants had said.

### 2.5. Ethical Consideration

Permission was obtained from the Gauteng Department of Health and selected facilities managers. Ethical clearance was obtained from the University of Venda research ethics committee and the Ekurhuleni Health District Research Committee. Participants were informed about the purpose of the study. Participants were encouraged to participate without any coercion. Throughout the study, participants’ rights and confidentiality were maintained. Informed verbal consent was obtained from each interviewed participant. Participants were informed that participation is voluntary, and they can withdraw at any time and that no remunerations were to be provided for being part of the study. To ensure anonymity, codes were used instead of names.

### 2.6. Data Analysis

All interviews were transcribed word-for-word and translated into English to ensure accuracy. The data analysis was performed using thematic analysis to identify patterns and key themes across participants’ responses. The transcripts were re-read, and the audio recordings were listened to ensure no essential information was lost or misinterpreted in the transfer. During the reading process, important topics were coded, and parallel codes were merged to create themes. One main theme and five sub-themes emerged. A literature review enabled contextualization of the study findings.

## 3. Results

### 3.1. Demographic Information of Participants

This section presents the demographic information of the PBFW who participated in the study (Table 1 below). The demographic information comprises age, marital status, nationality, educational level, employment status, and religion. The findings show that most participants were pregnant. The average age was between 25 and 38 years, and the median age was 27 years; 44% of the participants had a secondary level of education. 60% of the participants were unemployed, and 72% were single. All the participants were Christians. Table 1 outlines the participants’ demographic information.

The study findings are discussed in Table 2 below. The qualitative findings are presented in a narrative arrangement, verbatim extracts of the participants are presented, and relevant literature to support the findings is described. As a result, four main themes, each with associated sub-themes, highlighting both barriers and strategies to improve service utilization of EMTCT services, emerged during analysis of data based on the in-depth interviews conducted with 25 participants. These included barriers associated with EMTCT services utilization, facility-based strategies to improve EMTCT service uptake, community support for enhancing EMTCT engagement, and the role of partner support in service utilization. Participants’ quotes are used to support each theme on the table below.

### 3.2. Theme 1: Barriers Associated with EMTCT Services Utilization

The findings of the study indicate that the majority of participants indicated that they did not visit healthcare facilities for bookings during the early trimester of pregnancy. This has a negative impact on the health of the fetus or baby when HIV status is discovered in the late stages of pregnancy. This was supported by the following subthemes:

#### 3.2.1. Sub-Theme 1: Adequate Knowledge of EMTCT Services

The findings of the study indicate that most of the participants had insufficient knowledge of EMTCT services when they were pregnant. Lack of understanding regarding HIV status was observed through participants’ body gestures, and emotions such as crying, other patients emphasized the issues of witchcraft as an excuse for denial. Others stated that their sexual partners did not know about their HIV status; hence, they were afraid of taking ARVs in front of their partners. The findings of the study further indicate that most of the women opt to book their antenatal clinic (ANC) late in their pregnancy because they thought early ANC booking was for women who are living with HIV. Inadequate knowledge of EMTCT services has a negative impact on the uptake and retention.

**Participant 04, a pregnant 34-year-old woman,** has supported this by asserting that “I had other children before this pregnancy, thus why I have decided to sit at home, because with the previous pregnancy, I was HIV negative”.

**Participant 6, a pregnant 30-year-old woman,** said, “I didn’t have enough information or knowledge about EMTCT services because I thought that other children are HIV-negative, even this one would be negative. I had other children before this pregnancy, thus why I decided to sit at home, because with the previous pregnancy, I was HIV negative”.

**Participant 08, a pregnant 30-year-old woman,** stated that, “I booked ANC late in pregnancy because I took advantage of that I am HIV negative, and I didn’t see the need for me to book for ANC early. When I came to book, and they told me that I was sick, I couldn’t believe it.”

#### 3.2.2. Sub-Theme 2: Abusive Relationship (Gender-Based Violence)

The finding revealed that most women are in an abusive relationship, experiencing gender-based violence. Gender-based violence affects the utilization of EMTCT services because women would want to please their partners. This results in women not complying well with their treatment and not disclosing their HIV status to their partners due to fear of being beaten or divorced.

**Participant 7, a lactating 24-year-old woman,** stated that, “when my partner found out that I was HIV positive, taking treatment, he started beating me, he told me to stop taking medication, and he took my tablets and hid them away. After two months of not taking medication, he was arrested, and the other day, when I was cleaning the room, that is when I found my two containers of medication under the bed”.

**Participant 17, a pregnant 33-year-old woman,** said, “I haven’t disclosed my status to my partner because he is abusive, and that made it difficult for me to take treatment the right way”.

**Participant 5, a lactating 27-year-old woman,** said, “I am afraid that he might beat me or leave me because he is the one supporting me”.

#### 3.2.3. Sub-Theme 3: Denial State Due to Shyness, Hurt and Shock

The study results revealed that the majority of PBFW fail to accept their HIV status because of hurt or shock. Denial of either HIV status or pregnancy has a negative impact on the utilization of EMTCT service because most women will not comply with their clinic visit or adhere to their ART.

**Participant 10, a pregnant 30-year-old woman, said that** “when I first found out about my HIV status, I was hurt, but I had no one to blame because I was not careful enough. I could not take my treatment well at the beginning because I was still in denial”.

**Participant 14, a lactating 32-year-old woman, said**, “I was still in denial, I figured it was going to hurt me seeing my kid’s taking ART, so I decided to do the right things”.

**Participant 1, a pregnant 38-year-old woman,** said that “It is because I couldn’t believe that I was living with HIV, because after testing and they told me that I am diagnosed with HIV, I thought that they were lying to me. It was because my partner is not taking any treatment, so I thought maybe they have bewitched me, and I will be healed in no time after some religious consultations”.

#### 3.2.4. Sub-Theme 4: Socioeconomic Status of PBFW

Study results revealed that the majority of PBFW are financially dependent on their partners. This has shown that women who are more dependent on their partners cannot decide for themselves, as they depend on their partners. Socioeconomic status dependency affects the utilization of EMTCT services.

**Participant 5, a lactating 27-year-old woman,** said, “I am afraid that he might beat me or leave me because he is the one supporting me”.

**Participant 7, a lactating 24-year-old woman,** said, “I am now able to take medication because he is arrested, and I didn’t want to go against his will because he is the breadwinner in the house and would beat me if I disobey him. He is the one who brought me here to South Africa.’

### 3.3. Theme 2: Facility Strategies to Improve Utilization of EMTCT Services

Participants proposed facility-based strategies to improve the utilization of EMTCT services among PBFW in Gauteng Province, which are reflected in the following sub-themes.

#### 3.3.1. Sub-Theme 1: Treatment Navigation Model (TNM)

TNM is a patient-centered approach designed to improve healthcare access, adherence, and continuity of care [14]. A trained treatment navigator, a healthcare worker, counselor, or lay worker provides individualized guidance to patients, helping them overcome barriers such as a lack of information, stigma, or difficulty in accessing services. Participants appreciated telephone tracing to remind them of appointments and the collection of appropriate medications. They further stated that the treatment navigation model offers them support throughout the pregnancy and lactation periods, which promotes their EMTCT uptake and adherence. However, other participants said that they did not want to be called via telephone because they had to respect the POPIA Act and because it added stress when they were reminded of appointments.

**Participant 2, a 24-year-old lactating woman,** said, “Telephonic tracing to remind us about the clinic appointment also assists because if it wasn’t for the call that I received to come to the clinic, I wouldn’t have been here. Also, those community healthcare workers who check children in the community assist in providing information because when they found out that I was mixed feeding the baby, they told me that I shouldn’t mix feed the baby but only breastfeed the baby for six months and introduce feed after six months.”

#### 3.3.2. Sub-Theme 2: Health Education on EMTCT Services in the Facility

The participants believed that health education is crucial for educating patients attending EMTCT in healthcare facilities. This education alerts patients to test for HIV and know the results so that they can start ART treatment to prevent transmission. Furthermore, health education assists them in addressing barriers they come across, promotes healthy behavior, and empowers them to make the right decisions. It further reduces misconceptions about HIV and EMTCT services and encourages women to utilize EMTCT services in time and remain in care.

**Participant 4, a pregnant, 32-year-old woman,** reported that “Health education on EMTCT services, including education on transmission from mother-to-child, can help while we are sitting in the queue to get more information.”

**Participant 7, a 24-year-old lactating woman,** said, “I think educating women and men about HIV transmission from mother-to-child will help because most people do not have enough knowledge about EMTCT within the community.”

### 3.4. Theme 3: Community Support to Improve Utilization of EMTCT Services

Community support was acknowledged by PBFW as a facilitator to improve their access to EMTCT services. Two sub-themes emerged reflected below.

#### 3.4.1. Sub-Theme 1: Use of Community Healthcare Workers for Health Promotion in the Community/Home Visits

Participants reported that CHWs assisted in educating and providing information on the importance of EMTCT to pregnant women in the community. This encourages pregnant women to visit healthcare facilities in time to test for HIV. CHWs play a significant role in reaching PBFW and addressing health disparities. CHWs improve the utilization of EMTCT services such as HIV testing, early antenatal care booking, and adherence to antiretroviral treatment while offering support to women throughout their pregnancy and lactation period. Furthermore, participants confirmed that CHWs assisted them in addressing concerns in a supportive environment and promoted a reduction in the MTCT of HIV.

**Participant 6, a 36-year-old pregnant woman,** said, “Most people in the community do not have enough knowledge about EMTCT services. So, community healthcare workers should educate people in the community, and women living with HIV should disclose their status to their partners to improve treatment adherence.”

**Participant 8, a 30-year-old pregnant woman,** said, “Community awareness and involvement, where the community will be educated about EMTCT services and other infections. In addition, clinics should encourage every patient who visits the clinic to test for HIV at every visit they come to the clinic as a compulsory test. The other thing that can help the community get more information is through mobile clinics, where they also offer HIV testing, because most people get sick, but they don’t want to come to the clinic.”

#### 3.4.2. Sub-Theme 2: Establishment of Peer-to-Peer Support Groups

Participants confirmed that peer-to-peer support groups during lactation periods assisted them in utilizing EMTCT services. They considered peer-to-peer support groups a form of social support with other women who shared the same experience; it improved treatment adherence, reduced stigma associated with HIV and promoted remaining in care. Furthermore, the support groups assisted women in addressing their fears and concerns and offered them an opportunity to learn more about EMTCT services, feeding options, and other sexual reproductive health services. Peer-to-peer support groups encouraged women to stay engaged in care and promoted safeguards against the transmission of HIV from mother to child. They also enhanced knowledge about HIV and EMTCT among PBFW.

**Participant 4, a pregnant 32-year-old woman,** said, “Peer-to-peer support groups with other women where we can share our challenges because most women don’t take their treatment because they fear disclosing to their partner because they are afraid that their partner might leave them.”

**Participant 13, a pregnant 20-year-old woman,** said, “I think learning from other people who have been through it will be better, and now, because there is an ART treatment, it is much better as compared to the olden days”.

### 3.5. Partner Support to Improve Utilization of EMTCT Services

Participants have described that their partners are either influencing or discouraging their utilization of EMTCT services. one sub-theme emerged.

#### Sub-Theme 5: Male Involvement/Partner Support

Participants agreed that the involvement of both partners in visiting clinics during the pregnancy provided elements of care, support, and HTS for both partners. When men are actively engaged in supporting their female partners during pregnancy, childbirth, and breastfeeding, rates of EMTCT utilization tend to increase. This encourages partners to adhere to the ART and EMTCT services. It further facilitates the disclosure of HIV status and the intake of ART at home without fear. Moreover, male involvement in EMTCT utilization helps reduce the stigma and discrimination associated with HIV and makes it easier for women to access care. CHWs should increase awareness programs to educate community members on the importance of HIV testing, regardless of whether they are pregnant. These programs assist in the early detection of pregnancy and HIV. They further provide education in terms of condom utilization, the importance of ART adherence, feeding options, and safe conception.

**Participant 11, a pregnant, 23-year-old woman**, said, “I think disclosure to partner and encouraging partner testing might assist most women to comply with the treatment and clinic visits without any fear.”

**Participant 22, a lactating, 27-year-old woman**, said, “I think they should encourage men to test because they are the ones who don’t want to come to the clinic, and they are the ones who give us problems.”

## 4. Discussion

This study explored barriers associated with the utilization of EMTCT services and identified several strategies that PBFW perceive as helpful in improving the use of EMTCT services. The study showed that despite the implementation of EMTCT services, there are still barriers that affect the goal of EMTCT services. The study has revealed that there is a lack of adequate knowledge about EMTCT services among PBFW, including the importance of early antenatal booking, consistent ART throughout pregnancy and breastfeeding, and the risks of vertical HIV transmission. similar findings were identified from the study conducted in Uganda indicating that inadequate knowledge about EMTCT services was associated with delayed clinic visit [15]. Similarly, a South African study reported that inadequate information about the risk of vertical transmission of HIV and the benefits of EMTCT services affects the effectiveness of EMTCT program [16]. Some participants revealed that cultural belief about HIV influenced their ART uptake. One participant believed she had been “bewitched” and would be healed through religious consultations rather than through ART. These types of belief influence a health seeking behavior among PBFW in South Africa [17]. The study has revealed that many PBFW have experienced denial after learning that they are living with HIV, which has affected ART uptake. These findings were supported by studies that have revealed that denial of HIV status and stigma result in poor treatment adherence and poor retention in EMTCT services [18,19]. gender inequality emerged, several PBFW have reported abuse, coercive control and fear of losing financial support and disclosing their HIV status. these findings were supported by studies from Southern and Eastern Africa, which shows that women who experience intimate partner violence are less likely to access EMTCT services [20,21]. A study conducted in South Africa revealed that delayed HIV disclosure was associated with fear of partner reaction among PBFW. Many women are more dependable on their partners for financial support restricting their decision power making it hard to adhere to ART and EMTCT services. Finding from the studies supported the by highlighting that women who are economically dependent tend to book ANC early, adhere to ART throughout pregnancy and breastfeeding [22,23]. Several studies have revealed that unemployment or economic dependence contribute to poor treatment adherence and poor retention in care [24,25,26].

Participants have proposed several strategies to improve utilization of EMTCT services, facility strategies including treatment navigation, health education and community strategies including use of community healthcare worker involvement, establishment of peer-to-peer support, and male partner support. The treatment navigation model, which includes appointment reminders by phone and SMS, has a positive influence on motivating women to use EMTCT services. This aligns with the findings of Adhiambo et al. [27], who reported that navigation calls and messages foster trust, provide personalized support, and reinforce adherence. They further reported that treatment navigation provides additional information about EMTCT, as well as appointment reminders, which encourage women to use these services.

Health education about EMTCT services, delivered both at facilities and within communities, empowered women to make informed decisions about their care. This was supported by [4,28] highlighting that education about EMTCT and its benefits improves knowledge while promoting service utilization. Overall, it is critical to give thorough and culturally relevant education and counseling to women and their families. This can include information about the necessity of EMTCT services, how to prevent mother-to-child transmission, and what support resources are available. Empowering women with knowledge and resources allows them to make informed decisions [4].

Engaging community healthcare workers and other stakeholders, such as community leaders, clinic committees, and family members, boosts awareness, service uptake, and retention [29,30,31,32]. Structured community engagement, including home visits and information dissemination, helps bridge gaps between health facilities and local communities. These findings support the need to formalize the role of community health workers and provide training that strengthens their ability to effectively promote EMTCT services.

Studies have found that women tend to be economically dependent on their partners, and without their partners’ approval, they do not seek medical help [33,34,35]. This is aligned with the findings outlined in [24,25], which reported that male partner involvement and peer-to-peer support were important facilitators. Male engagement encourages HIV disclosure, adherence, and support from partners and family [33,34,35,36]. These findings were reinforced by the Mentor Mothers Zithulele study, which was conducted in South Africa and indicated that peer-to-peer assistance plays a significant role in lowering HIV MTCT through the provision of health education and social support [37]. Studies suggest that the peer-to-peer support model provides considerable improvement in EMTCT service utilization and maternal conduct [32,38,39,40]. Furthermore, training healthcare workers to integrate peer support into their services can enhance the overall effectiveness of EMTCT programs. Overall, this study demonstrates that multi-level strategies integrating treatment navigation model, health education, community engagement, peer support, and male involvement are essential to improve the utilization of EMTCT services and achieve an HIV-free generation.

### Limitations

This study was conducted in one province of South Africa and focused exclusively on PBFW, which limits the generalizability of the findings to other regions or populations. The qualitative design and relatively small sample size also restrict the generalization of these results. Selection bias may have occurred because participants were only recruited among women utilizing EMTCT services, thus omitting those facing barriers to care. Furthermore, social desirability bias may have influenced the participants’ responses, as they may have given answers that they believed were expected or acceptable, particularly regarding sensitive themes such as HIV declaration, partner engagement, and adherence to EMTCT services.

## 5. Conclusions

The study has provided a better understanding of the barriers affecting utilization of EMTCT services and facilitators encouraging PBFW to utilize these services. Utilization of EMTCT services was affected by knowledge gaps, fear of partner violence, denial and financial dependence. PBFW expressed treatment navigation models such as phone calls or SMS reminders, health education at facilities, support from community health workers, establishment of peer-to-peer support groups, and partner support as facilitators to improve utilization of EMTCT services. To enhance the utilization of EMTCT services and reduce the risk of vertical transmission of HIV, interventions should address multiple levels of care and support. Policymakers and healthcare providers should incorporate treatment navigation, community-based education, and peer-to-peer support into routine EMTCT programs. Engaging male partners and providing culturally sensitive training for healthcare workers are crucial for promoting adherence, reducing stigma, and creating a supportive environment for women. Collaboration between governments, healthcare facilities, and community organizations is essential to ensure that services are not only accessible but also welcoming and responsive to the real needs of PBFW. By putting these strategies into practice, EMTCT programs can be strengthened, maternal and child health outcomes improved, and progress toward an AIDS-free generation accelerated.

## Figures and Tables

**Table 1 nursrep-15-00318-t001:** The demographic profile of pregnant and lactating mothers.

Participant #	Preg/Lactating	Age	Nationality	Marital Status	Education	Employment	Religion
*N* = 25	Pregnant: 56% Lactating: 44%	Age range: 20–38 years Median age: 27 years Mean age: 28.2 years In their twenties (20–29 years): 15 participants (60%) In their thirties (30–39 years): 10 participants (40%)	South Africans: 100% (*n* = 25)	Single: *n* = 18 (72%) Married: 5 (20%) Widow: *n* =1 (4%) Divorced: *n* =1 (4%)	Secondary: *n* = 11 (44%) Tertiary: *n* =8 (32%) No education: *n* = 6 (24%)	Unemployed: *n* = 15 (60%) Employed: *n* = 9 (36%) Self-employed: *n* = 1 (4%)	Christianity: *n* = 25 (100%)

**Table 2 nursrep-15-00318-t002:** Barriers and Strategies in EMTCT Services Utilization.

Main Theme	Sub-Theme	Participant Quotes
1. Barriers associated with EMTCT Services Utilization	Inadequate Knowledge of EMTCT	Participant 4, 34 years, *“I had other children before this pregnancy, thus why I have decided to sit at home…”. “When I found that I was pregnant I decided that I would come to the clinic when I was 4 months pregnant.”* *Participant 6, 30 years, “I didn’t have enough information or knowledge about EMTCT services because I thought that other children are HIV-negative, even this one would be negative.”.*
	Gender-Based Violence	Participant 17, 33 years, *“I haven’t disclosed my status to my partner because he is abusive and that made it difficult for me to take treatment.” Participant 5, 27 years, “I am afraid that he might beat me or leave me because he is the one supporting me.”*
	Denial, Shyness, Shock	Participant 10, 30 years, *“When I first found out about my HIV status, I was hurt... I could not take my treatment well.” Participant 14, 32 years, “I was still in denial, I figured it was going to hurt me seeing my kids taking ART…” Participant 1, 38 years, “I thought maybe they have bewitched me, and I will be healed with no time after some religious consultations.”*
	Socioeconomic Status	Participant 5, 27 years, *“I am afraid that he might beat me or leave me because he is the one supporting me.” Participant 7, 24 years, “He is the one who brought me here in South Africa.”*
2. Facility strategies to Improve utilization EMTCT Services	Treatment Navigation Model	Participant 2, 24 years, *“If it wasn’t for the call that I received to come to the clinic, I wouldn’t have been here. CHWs who check children in the community assist in providing information...”*
	Health Education	Participant 4, 32 years, *“Health education on EMTCT services, including education on transmission from mother to child, can help while we are sitting in the queue...” Participant 7, 24 years, “Educating women and men about HIV transmission from mother to child will help.”*
3. Community support to improve utilization of EMTCT services	Use of community health for health promotion in community/home visits	Participant 5, 27 years, *“Community healthcare workers should educate women in the community about EMTCT services.”* Participant 6, 36 years, *“CHWs should educate people and women living with HIV should disclose their status to improve adherence.”*
	Establishing peer-to-peer support groups	Participant 4, 32 years, *“Peer-to-peer support groups with other women where we can share our challenges...”* Participant 13, 20 years, *“Learning from other people who have been through it will be better...”*
4. Partner support to improve utilization of EMTCT services.	Male Involvement/partner Support	Participant 11, 23 years, *“Disclosure to partner and encouraging partner testing might assist most women to comply.”* Participant 22, 27 years, *“They should force men to test because they are the ones who give us problems.”*

## Data Availability

The data supporting this study’s findings are available from the corresponding author upon reasonable request. However, an MOU must be signed between the two parties, including the journal and authors, to ensure that ethical procedures are followed in all aspects of the study.

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
