# Peer review of "Barriers and Facilitators to the Elimination of Mother-to-Child Transmission Services Among Pregnant and Breastfeeding Women in Gauteng Province, South Africa"

_nursrep, 2025, doi:10.3390/nursrep15090318_

Round 1

Reviewer 1 Report

Comments and Suggestions for Authors

Overall, the study has potential but needs significant revision in structure, clarity, terminology, thematic analysis, and alignment between title, objectives, methodology, and results as per the comments in the attachmented review report. The study results could be expanded by incorporating additional subthemes under the three main themes proposed in the review. This would provide deeper insights and a broader range of perspectives from pregnant and breastfeeding women regarding the utilization of EMTCT services 

Comments on the Quality of English Language

The manuscript should undergo professional editing to address the numerous grammatical and semantic issues throughout the text.

Author Response

Responses to reviewers comments are attached

Reviewer 2 Report

Comments and Suggestions for Authors

Reviewer’s Comment.

Your manuscript has a good introduction; however, you should consider addressing the following:

  1. Refer to 2.5. Data quality:

-Is this section necessary in your manuscript? How relevant is the data quality section? It is not very clear. You mentioned that “the researchers contacted participants during the appointment session, information session, and data collection.” What was the purpose of contacting them?

-Also, what is referential adequacy? – What do you mean by saying “referential adequacy was achieved by taking notes to provide a suitable record of the interview……”?

-You mentioned that one researcher conducted the study/or did the analysis, as you have described throughout the methods. So, why are you saying researchers contacted participants in the data quality section? (Refer to page 2. Line 139.)

-The data quality section sounds repetitive and reads like a text. Consider revising section 2.5 or removing it from the methods.

  1. Refer to 3. Results:

-In the methods, you stated that “one main theme and five sub-themes emerged” from the data analysis. However, on page 4 in the results, line 151, you said, “….. two-sub themes emerged during the analysis of data…” Fix that error.

-Consider moving Figure 1 to the data analysis section. I think it fits better in the analysis section than in the results.

-Also, you may consider presenting the theme and sub-themes in Figure 1 in table format. Consider creating a simple table (3 by 6 table) like the table below.

Main theme

Sub-themes

Quotes

Insert main theme

insert subtheme 1

Insert one or two supporting direct quotes from the interviews

Insert sub-theme 2

ii

Insert sub-theme 3

ii

Insert sub-theme 4

Inset sub-theme 5

-Please collapse the table 1: demographics profile… You create an 8 by 2 table.

-For the age, you can report only the range, the median age, and the mean age. You can also add the percentage/number of those who were in their twenties and those in their thirties. Please see an example of a collapsed table below.

Participant #

Preg/Lactating

Age

Nationality

Marital Status

Education

Employment

Religion

N=25

Pregnant:56%

Lactating: 44%

Age range:

Medium age:

Mean age:

South Africans:100% (n=25)

-Refer to 3.1.1. First, consider defining the Treatment Navigation Model (TNM) before presenting participants' perspectives on it, to help readers who may not be familiar with the TNM.

-Replace the word “Respondents” with “Participants.” Usually, “respondent” is used in quantitative studies, while “Participant” is common in qualitative studies. Refer to page 5, line 173; page 7, line 254…….

-Your direct quotes for each sub-theme should not exceed three. Supporting each sub-theme with two direct quotes is usually enough. Including more than two quotes can make the reading dull unless they are integrated into the findings’ narratives paragraph by paragraph. For example, in Sub-theme 2 of the Results, you included five direct quotes. Please consider reducing this to two, as more than three is unnecessary. Also, remove the text you have bolded from the participants' information in the direct quotes.

 3. Refer to 4. Discussion

-Page 8, lines 296-299, you stated that “According to Kim et al.., Marcose et al..,Kinuthia et al.,& Mukomafhedzi et al, why are you referring to several studies in that order? Why not put it this way? ‘Previous studies have revealed that involvement of various stakeholders such as community leaders… (and then cite those studies at the end of the sentence).

-Please consider limiting the use of “According to” in your discussion. It makes it repetitive. Refer to page 8, the sentences starting on lines 296, 306, and 312.

-On the limitations of the study, is only the location an issue for the generalization of the findings? What about the fact that it was a qualitative study, and the sample size was also small?

Comments on the Quality of English Language

It will require editing. 

Author Response

Responses for reviewers comments attached

Reviewer 3 Report

Comments and Suggestions for Authors

Congratulations to the authors on their work addressing women’s utilization of elimination of mother-to-child transmission (eMTCT) services in Gauteng Province, South Africa. This is an important topic in the context of HIV elimination efforts, and the manuscript highlights strategies that could improve service uptake among pregnant and lactating women.

However, substantial effort is required to improve clarity, organization, and adherence to qualitative reporting standards. 

Below are specific comments and suggestions to improve the manuscript:

General: 

  • The flow of the manuscript can be improved. Replace passive voice with active voice throughout for more concise scientific writing.

  • Replace “HIV-positive” with “people with HIV” throughout the manuscript. 

Minor Editorial

Line 6–11: Consider removing “Affiliations” before listing of author affiliations.

Line 49: “UNAIDS believes”?? → “UNAIDS reports/estimates.”

Line 50–52: Clarify if ART reduces MTCT risk to <1% or reduces risk by 1%.
Line 59: ART = antiretroviral therapy.

Rephrase ethically sensitive statements (e.g., “force men to test”) to “encourage or promote male testing.”

Abstract

  • Line 19: Consider rewording the statement implying a direct link between poor eMTCT utilization and maternal/neonatal deaths; it should emphasize indirect contribution to adverse outcomes.

  • Include study setting (City of Ekurhuleni, Gauteng Province) in Objectives or Methods.

  • The consistent use of “researcher” suggests the study was conducted by a single person instead of a study team. Consider improving the wording. 

Introduction

The introduction does not set up the study logically. A suggested structure:

  1. Global context – HIV and MTCT burden, eMTCT as a global priority.

  2. National context – South Africa’s HIV burden, eMTCT programs, and Gauteng’s relevance.

  3. Problem & knowledge gap – Persisting underutilization and lack of evidence on women’s perceived strategies.

  4. Rationale & study aim

Methods

  • The study period is missing.

  • Clarify whether the semi-structured interview guide was piloted; if not, justify.

  • Setting: Describe how Ekurhuleni demographics compare with Gauteng and national data to contextualize findings.

  • Clarify the literature review approach; if it was not a formal method, remove it from Methods.

  • Mention whether multiple coders or inter-rater checks were done for thematic analysis.

Results

  • Present results without interpretation; avoid discussion in this section.

  • Table 1 should include summary statistics (median age, % unemployed, etc.).

Discussion

  • Current discussion largely repeats results. Improve by synthesizing findings with existing literature, highlighting novel contributions, and expanding policy and programmatic implications for eMTCT in South Africa.

  • Limitations: Is there a selection bias in the approach that was used to gather study participants, or potential social desirability bias from interviews? These are not mentioned

  • Make the conclusion more action-oriented, and emphasize implementation and policy recommendations rather than restating results.

Comments on the Quality of English Language

The flow of the English language can be improved. Replace passive voice with active voice throughout for more concise scientific writing.

Author Response

Responses to reviewers' comments is attached

Round 2

Reviewer 1 Report

Comments and Suggestions for Authors

The authors have made significant improvements, including:

The title has been revised to be more specific and aligned with the study's scope, and the terminology has been updated to reflect current standards, such as using "Pregnant and Breastfeeding Women" (PBFW) and "vertical transmission" where appropriate.

The abstract, results, and discussion sections have been completely restructured, which has significantly improved the manuscript's flow and clarity. The results section now presents multiple main themes, and the discussion provides a robust comparison of the findings with existing literature. The ethical consideration part was inserted as well to my satisfaction.

All inconsistencies in abbreviations have been corrected, and the reference list has been standardized. The authors have also ensured the participant quotes are balanced and relevant to each subtheme.

 The revisions have substantially enhanced the quality of the paper. 

As requested in my last review, if each word of the title can be capitalized unless it is the requirement of the journal to keep it as such

Reviewer 2 Report

Comments and Suggestions for Authors

The authors have responded to my comments. This is an improved version.